# Massive computational acceleration by using neural networks to emulate mechanism-based biological models

Shangying Wang[1], Kai Fan[2], Nan Luo[1], Yangxiaolu Cao[1], Feilun Wu [1], Carolyn Zhang[1], Katherine A. Heller[2] & Lingchong You [1,3,4]*

For many biological applications, exploration of the massive parametric space of a mechanism-based model can impose a prohibitive computational demand. To overcome this limitation, we present a framework to improve computational efficiency by orders of magnitude. The key concept is to train a neural network using a limited number of simulations generated by a mechanistic model. This number is small enough such that the simulations can be completed in a short time frame but large enough to enable reliable training. The trained neural network can then be used to explore a much larger parametric space. We demonstrate this notion by training neural networks to predict pattern formation and stochastic gene expression. We further demonstrate that using an ensemble of neural networks enables the self-contained evaluation of the quality of each prediction. Our work can be a platform for fast parametric space screening of biological models with user defined objectives.

[1] Department of Biomedical Engineering, Duke University, Durham, NC 27708, USA. [2] Department of Statistical Science, Duke University, Durham, NC 27708, USA. [3] Center for Genomic and Computational Biology, Duke University, Durham, NC 27708, USA. [4] Department of Molecular Genetics and Microbiology, Duke University School of Medicine, Durham, NC 27708, USA. *email: you@duke.edu

Mathematical modeling has become increasingly adopted in analyzing the dynamics of biological systems at diverse length- and time-scales[1–6]. In each case, a model is typically formulated to account for the biological processes underlying the system dynamics of interest. When analyzing a gene circuit, the corresponding model often entails description of the gene expression; for a metabolic pathway, the corresponding model may describe the constituent enzymatic reactions; for an ecosystem, the corresponding model would describe growth, death, and movement of individual populations, which could in turn be influenced by other populations. We call these models mechanism-based models.

Mechanism-based models are useful for testing our understanding of the systems of interest[7–14]. For instance, modeling has been used to examine of the network motifs or the parameter sets able to generate oscillations[15,16] or spatial patterns[17], or the noise characteristics of signaling networks[18–21]. They may also serve as the foundation for practical applications, such as designing treatments of diseases[22–24] and interpreting the pharmacokinetics of drugs[25–27]. Many mechanism-based models cannot be solved analytically and have to be analyzed by numerical methods. This situation is particularly true for models dealing with spatial or stochastic dynamics. While numerical simulations are typically more efficient than experiments, they can still become computationally prohibitive for certain biological questions. For example, consider a model with 10 parameters. To examine six values per parameter, there will be $6^{10}$ parameter combinations. If each simulation takes 5 min, which is typical for a partial differential equation (PDE) model, the screening would require 575 years to finish. Many biological systems are much more complex. For each system, both the size of the parametric space and the time required to do each simulation would increase combinatorially with the system complexity. Thus, standard numerical simulations using mechanism-based models can face a prohibitive barrier for large-scale exploration of system behaviors.

Thanks to its ability to make predictions without a full mapping of the mechanistic details, deep learning has been used to emulate time-consuming model simulations[28–31]. To date, however, the predicted outputs are restricted in categorical labels or a set of discrete values. By contrast, deep learning has not been used to predict outputs consisting of continuous sequences of data (e.g., time series, spatial distributions, and probability density functions). We overcome this limitation by adopting a special type of deep learning network, the Long-Short-Term Memory (LSTM) network. For a pattern formation circuit, our approach leads to ~30,000-fold acceleration in computation with high prediction accuracy. We further develop a voting strategy, where multiple neural networks are trained in parallel, to assess and improve the reliability of predictions.

## Results

**The conceptual framework.** When numerically solving a mechanism-based dynamic model consisting of differential equations, the vast majority of the time is spent in the generation of time courses. For many biological questions, however, the main objective is to map the input parameters to specific outcomes, such as the ability to generate oscillations or spatial patterns[32–37]. For such applications, the time-consuming generation of time courses is a necessary evil.

The key to the use of the deep learning is to establish this mapping through training to bypass the generation of time courses, leading to a massive acceleration in predictions (Fig. 1). To do the learning, we use a small proportion of data generated by the mechanism-based model to train a neural network. The data generated by the mechanistic model need to be sufficiently

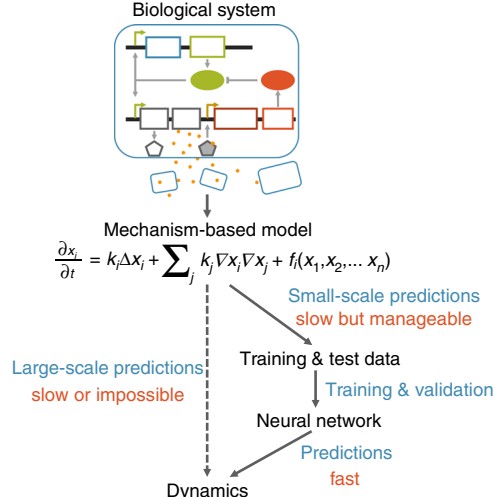

**Fig. 1** Using an artificial neural network to emulate a mechanism-based model. Here a hypothetic biological network and the corresponding mechanistic model are shown. The mechanistic model is used to generate a training data set, which is used to train a neural network. Depending on the specific mechanistic model, the trained neural network can be orders of magnitude faster, enabling exploration of a much larger parametric space of the system

large to ensure reliable training but small enough such that the data generation is computational feasible.

As a proof of principle, we first apply our approach to a well-defined model developed by Cao et al.[32] This PDE model describes pattern formation in *Escherichia coli* programed by a synthetic gene circuit (Methods and Supplementary Fig. 1a), accounting for cell growth and movement, intercellular signaling and circuit dynamics as well as transportation (Eq. 1). This model was previously used to capture the generation of characteristic core-ring patterns and to examine the scaling property of these patterns. Numerical simulations were used to explore the parametric space to seek parameter combinations able to generate scale-invariant patterns. Several months were needed to search through 18,231 parameter sets. Yet, these parameter sets only represent an extremely tiny fraction of the parametric space that the system can occupy. Thus, it is likely that these numerical simulations have not revealed the full capability of the system in terms of pattern formation. For example, it is unclear whether the system can generate more than two rings and how this can be achieved.

For this system, each input is a set of parameters (e.g., cell growth rate, cell motility, and kinetic parameters associated with gene expression); the output is the spatial distribution of a molecule. The mapping between the two is particularly suited for the use of an LSTM network. The LSTM network, a type of recurrent neural network (RNN), was proposed in 1997 to process outputs consisting of a continuous series of data[38]. It has demonstrated great potential in natural language processing and speech recognition as well as in other sequence-prediction applications[39].

The outputs of the model can vary drastically in the absolute scale. To improve the learning process, we break each output profile into two components: the peak value of each profile and the profile normalized with respect to the peak value. Our deep neural network consists of an input layer with inputs to be the parameters of mechanism-based model, connected to a fully connected layer, and the output layer consists of two types of outputs, one for predicting the logarithm of the peak value of the profile, directly connected to the fully connected layer, the other

for predicting the normalized profile, connected to LSTM cell arrays, which was fed by the output from fully connected layer. The detailed structure of the neural network is described in Methods and Supplementary Fig. 2.

**The neural network accurately predicts spatial distributions.** To train the neural network, we first used our PDE model to generate $10^5$ simulation results from random combinations of parameter values. Generating these data sets was manageable: it took 2 months on a cluster consisting of 400 nodes. We split the data sets into three groups: 80% for training, 10% for validation and 10% for testing. We used the root mean squared error (RMSE) to evaluate the difference between the data generated by PDE simulation and those generated by the neural network (also see Methods and Supplementary Table 4). Each output distribution was dissected into two components for prediction: the peak value, and the shape of the distribution (i.e. the distribution after being normalized with respect to the peak value).

The trained neural network is fast and highly accurate. For each set of parameters, the neural network on average enabled ~30,000-fold computational acceleration (see Methods), though the specific extent of acceleration will vary with specific models. The correlation between predicted values and PDE simulation results exhibits high $R^2$ values: 0.987 for peak value predictions and 0.998 for shape predictions (Fig. 2a). For most parameter sets, distributions generated by the neural network align nearly perfectly with those generated by numerical simulations (Fig. 2b, Supplementary Figs. 3 and 4). In general, the more complex the output distribution, the less accurate the prediction (Supplementary Table 4). This trend likely results from the uneven representation of different types of patterns in the training data

sets: the majority have no ring (42897 out of $10^5$) or have only one ring (55594 out of $10^5$); patterns with multiple rings are rare (1509 out of $10^5$ for 2 rings or more) (Supplementary Fig. 5).

To identify the minimum size of dataset needed for accurately making predictions, we trained deep LSTM network on different training dataset sizes. The RMSEs are calculated based on predictions of a fixed test dataset, which contains 20,000 samples. Figure 2c demonstrates how the RMSEs of distributional data and peak values decrease with the increase of training data size. Since the x-axis is log-scaled, when the dataset size is beyond $10^4$, the rate of error reduction becomes asymptotically smaller. When the data size is $10^5$, the RMSE value decreased below a preset threshold (0.3), when we deemed a prediction to be accurate. In general, depending on specific models and the acceptable tolerance of errors, the threshold can be set differently, which could require different data sizes for effective training. This training dataset size is manageable and results in sufficient accuracy for our analysis. Based on error tolerance and numerical data generation efficiency, one can choose the desired dataset size for training the neural network. With an ensemble of deep neural networks, which will be described in the next section, the errors can be further reduced without increasing the dataset size.

**The neural network predicts novel patterns.** We use the trained deep LSTM network to screen through the parametric space. It takes around 12 days to screen through $10^8$ combination of parametric sets, which would need thousands of years if generated with PDE simulations. We find 1284 three-ring pattern distributions, including novel patterns not present in the training sets (Fig. 3). These are genuinely novel three-ring patterns found in this screening process, which are not in the training dataset.

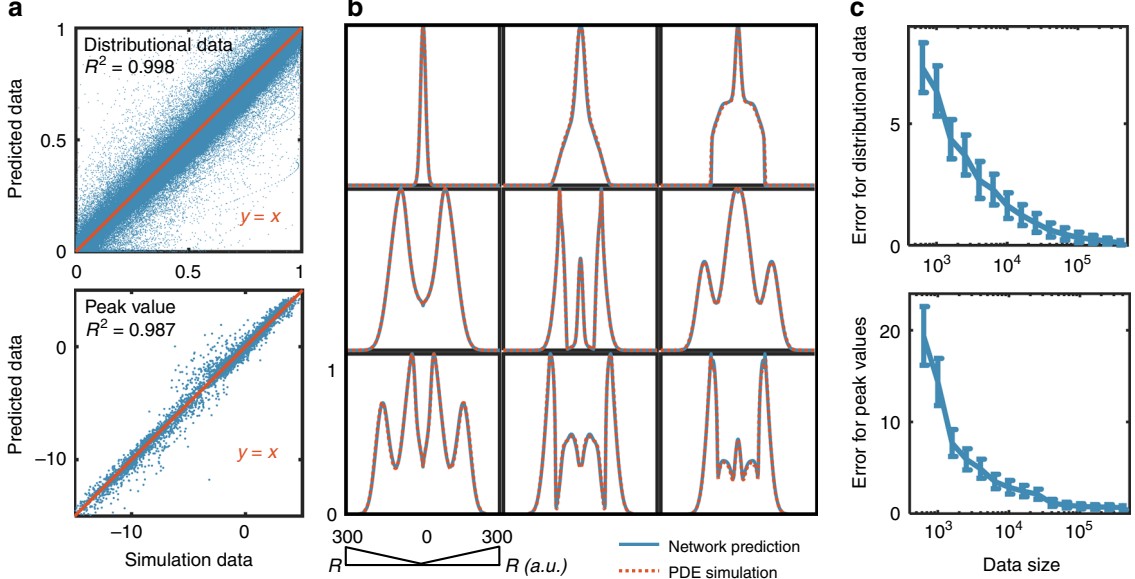

**Fig. 2** Neural network training and performance. We generated $10^5$ simulated spatial distributions using our partial differential equation (PDE) model and split the data into three groups: 80% for training, 10% for validation and 10% for test. We used root mean squared errors (RMSEs) to evaluate the differences between data generated by the mechanism-based model and data generated by the neural network. **a** Accuracy of the trained neural network. The top panel shows the predicted distributions by the neural network plotted against the distributions generated by numerical simulations. The bottom panel shows the peak values predicted by the neural network plotted against the peak values generated by numerical simulations. Perfect alignment corresponds to the $y = x$ line. The test sample size is s (=10,000). Each spatial distribution consists of 501 discrete points; thus, the top panel consists of 5,010,000 points. **b** Representative distributions predicted by neural network from test dataset. Each blue line represents a predicted distribution using the trained neural network; the corresponding red dashed line represents that generated by a numerical simulation. Additional examples are shown in Supplementary Figs. 3 and 4. **c** Identifying the appropriate data size for reliable training. The top panel shows the RMSE between distributions generated by the neural network and the distributions generated by numerical simulations as a function of an increasing training data size. The bottom panel shows the RMSE of peak-value predictions as a function of an increasing training data size. The RMSEs are calculated based on predictions of a test dataset, which contains 20,000 samples

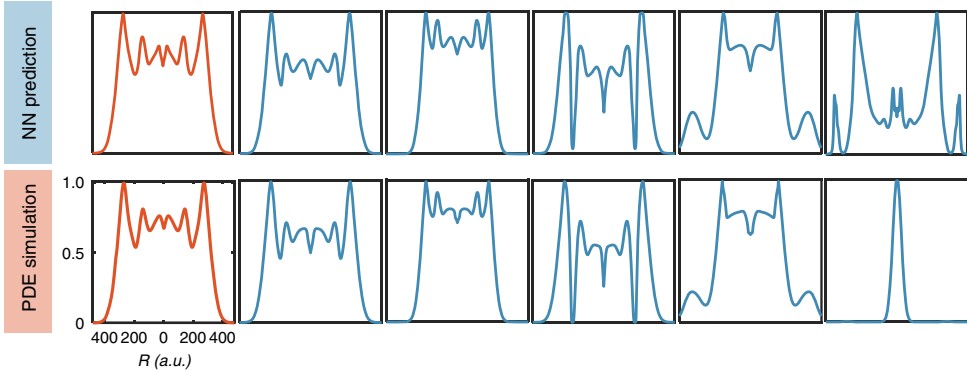

**Fig. 3** The trained neural network predicts novel patterns. We used the neural network to screen $10^8$ parameter combinations to search for three-ring patterns. We then used the mechanism-based model to test accuracy of predicted patterns. We tested 1284 three-ring patterns and the mean value of the RMSEs between neural network predicted distributions and PDE simulations is 0.079 and the standard deviation is 0.008. The distributions shown in red are from training data set. The other distributions are from the screening process (top) and the corresponding results generated by the mechanism-based model for validation (bottom). In four examples, the neural network predictions are validated. In one, the neural prediction is incorrect. The RMSE values of these distributions (from left to right) are: 0.0099, 0.015, 0.0097, 0.039, 0.031, and 0.41

We further tested these neural network predictions by numerical simulations using the PDE model. Only 81 out of 1284 (i.e., 6.31%) neural network predicted three-ring patterns exceeded a stringent RMSE threshold of 0.1 (Supplementary Table 3). The high reliability in predicting novel patterns indicates that the learning by the neural network is not limited to passive recollection of what the network has been trained with. Instead, the training has enabled the neural network to establish the genuine mapping between the input parameters and the system outputs, in a manner that is highly non-intuitive.

**Voting enables estimation of prediction accuracy**. Despite the extremely high accuracy in the predictive power of the trained neural network, it is never 100% correct. This apparent deficiency is the general property of neural networks. The lack of perfection in prediction raises a fundamental question: when dealing with a particular prediction, how do we know it is sufficiently reliable? Even if it were feasible, validating every prediction by simulation, as done in the last section (Fig. 3), would defeat the purpose of using deep learning. Therefore, it is critical to develop a metric to gauge the reliability of each prediction, without resorting to validation using the mechanism-based model.

The wisdom of crowds refers to the phenomenon in which the collective knowledge of a community is greater than the knowledge of any individual[40]. To this end, we developed a voting protocol, which relies on the training of several neural networks in parallel using the same set of training data. Even though these networks have the same architecture, the training process has an intrinsically stochastic component. Each network creates a map of virtual neurons and assigns random numerical values, or weights, to connections between them during the initialization process. If the network does not accurately predict a particular pattern, it will back-propagate the gradient of the error to each neuron and the weights would be updated to minimize the error in a new prediction. Even though the same rule is applied, each neuron is updated independently. With same training data, same architecture, the probability of getting exactly the same parameterized neural network is essentially zero. That is, each trained neural network is unique in terms of the parameterization of the network connections. Supplementary Figure 6a illustrated the differences of trainable variables (weights, bias) between two trained neural networks. Despite the difference in parameterization, the different neural networks

trained from the same data overall make similar predictions. We reasoned that this similarity can serve as the metric of the accuracy of the prediction. In particular, for a certain input parameter set, if all trained networks give very similar predictions, it is likely that these predictions are overall accurate. In contrast, for another input parameter set, if predictions from different networks diverge from each other, this divergence would suggest some or all of these predictions are not reliable. Given this reasoning, we could expect a positive correlation between the reliability of the prediction (in comparison to the correct prediction generated by the mechanism-based model) and the consistency between predictions generated by different neural networks.

To test this notion, we trained four neural networks. For each parameter combination in the testing set, we calculated the divergence between predictions by different neural networks, by using the RMSE. The final prediction is the one with the least average RMSE between all other predictions (Fig. 4a). We then calculated the divergence between the ensemble prediction and the correct profile. Indeed, the accuracy of the ensemble prediction is positively correlated with the consistency between different neural networks (Fig. 4b, Supplementary Fig. 6b). That is, if predictions by neural networks exhibit high consensus, the errors in the prediction are also low. Also, a side benefit of using multiple neural networks is that the ensemble prediction is in general more reliable than one by a single neural network. The average RMSE over the test dataset reduced from 0.0118 to 0.0066. This improved accuracy is expected and is the typical use of ensemble method[41]. We further tested the voting strategy by using ensembles of three or five neural networks (Supplementary Fig. 6b). In each case, the consensus between neural networks correlated with the accuracy of the ensemble prediction, which did not vary significantly with the number of neural networks used. In general, the voting strategy is highly scalable: the different neural network predictors can be trained in parallel, via different graphics processing unit (GPU) cores or even different servers on a computer cluster. Similarly, predictions can be made in parallel.

We screened through $10^8$ combination of parametric sets using the ensemble prediction method, where we discarded predictions with disagreement in predictions larger than 0.1. These neural network predictions reveal the general criterion for making complex patterns. For example, generation of three-ring patterns requires a large domain radius (R), large synthesis rate of

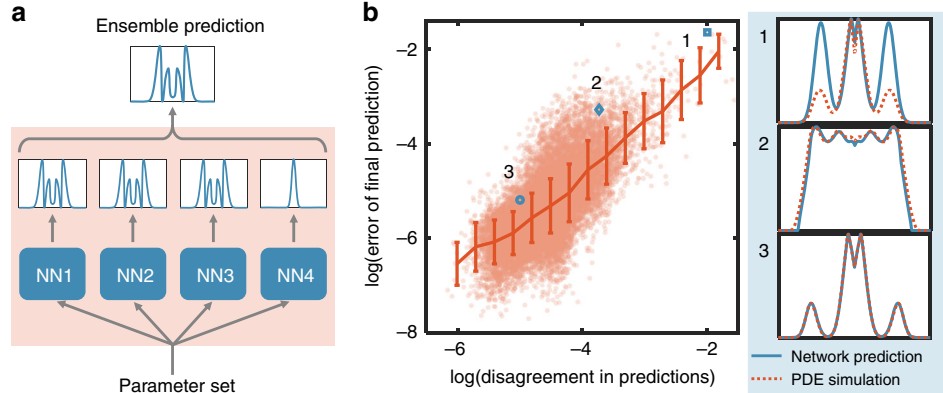

**Fig. 4** Ensemble predictions enable self-contained evaluation of the prediction accuracy. **a** Schematic plot of ensemble prediction. With each new parameter set (different combinations of parameters), we used several trained neural networks to predict the distribution independently. Though these networks have the same architecture, the training process has an intrinsically stochastic component due to random initialization and backpropagation. There might be multiple solutions for a certain output to be reached due to non-linearity. Despite the difference in parameterization of trainable variables (Supplementary Fig. 6a), the different neural networks trained from the same data overall make similar predictions. Based on these independent predictions, we can get a finalized prediction using a voting algorithm. **b** The disagreement in predictions (DP) is positively correlated with the error in prediction (EP). We calculated the disagreement in predictions (averaged RMSE between predictions from different neural networks) and error of final prediction (RMSE between final ensemble prediction and PDE simulation) for all samples in test data set (red dots). We then divided all data into 15 equally spaced bins (in log scale) and calculated the mean and standard deviation for each bin (red line). Error bar represents one standard error. The positive correlation suggests that the consensus between neural networks represents a self-contained metric for the reliability of each prediction. We showed three sample predictions with different degrees of accuracy: sample 1: DP = 0.14, EP = 0.19; sample 2: DP = 0.024, EP = 0.038; sample 3: DP = 0.0068, EP = 0.0056

T7RNAP ($\alpha_T$), small synthesis rate of T7 lysozyme ($\alpha_L$), small half activation constant of T7RNAP ($K_T$), small half activation distance for gene expression ($K_\varphi$) (Fig. 5a). Based on the analysis above, we want to further identify the correlation between $K_T$ and $\alpha_C$, $K_T$ and $\alpha_T$,$D$, and $\alpha_C$. For each of the screening, we vary two parameters of interest and fixed the rest to identify the relationship between parameters required to generate three-ring patterns. We collect $10^7$ instances and discard predictions with disagreement between ensemble predictions larger than 0.1 (Fig. 5b–d, Supplementary Fig. 7). We found that if the growth rate on agar ($\alpha_C$) is large, the domain radius ($D$) can be reduced. Additionally, there is a negative relationship between cell growth rate on agar ($\alpha_C$) and half activation constant of T7RNAP ($K_T$) (If approximating that they are inversely proportional, we can get the fitting with $R^2 = 0.94$. Supplementary Fig. 7a). We also found a linear correlation between half activation constant of T7RNAP ($K_T$), and synthesis rate of T7RNAP ($\alpha_T$) in order to generate three-ring patterns ($R^2 = 0.996$, Supplementary Fig. 7b). A key advantage is that machine-learning methods can sift through volumes of data to find patterns that would be missed otherwise. This provides significant insight in our experiments to find conditions that allow the formation of multiple rings, which could not be done using traditional simulation methods.

**High accuracy on predicting probability density functions**. Our framework is applicable to any dynamic model that generates a continuous series of each output. To illustrate this point, we apply the framework to the emulation of a stochastic model of the MYC/E2F pathway[42,43] (Supplementary Fig. 8a). This model consists of 10 stochastic differential equations and 24 trainable parameters (see Methods, supplementary Notes, and Supplementary Table 5). For each parameter set, repeated simulations lead to generation of the distribution of the levels of each molecule. With a sufficiently large number of simulations, this distribution converges to an approximately continuous curve (Supplementary Fig. 8b). As such, establishing the mapping between the parameter set and the corresponding output

distribution is an identical problem as prediction of the spatial patterns. Again, the trained neural network exhibits high accuracy in predicting the distributions of different molecules in the model (Supplementary Fig. 9).

## Discussion

Our results demonstrate the tremendous potential of deep learning in overcoming the computational bottleneck faced by many mechanistic-based models. The key to the massive acceleration in predictions is to bypass the generation of fine details of system dynamics but instead focus on an empirical mapping between input parameters to system outputs of interest using neural networks. This strategy contrasts with several previous studies, where neural networks have been adopted to facilitate numerically solving differential equations[44–52]. The massive acceleration enables extensive exploration of the system dynamics that is impossible by solely dependent on the mechanistic model (Fig. 5). Depending on the application context, this capability can facilitate the engineering of gene circuits or the optimization of experimental conditions to achieve specific target functions (e.g., generation of multiple rings from our circuit), or to elucidate how a biological system responds to environmental perturbations (e.g., drug treatments).

A major innovation of our approach is the combined use of the mechanistic model and the neural network (Fig. 1). The mechanistic model is used as a stepping stone for the latter by providing a sufficient data set for training and testing. This training set is extremely small compared with the possible parameter space. Given the relatively small training set, the remarkable performance of the neural network suggests that, for the models we tested, the landscape of the system outputs in the parametric space is sufficiently smooth. If so, a small training set is sufficient to reliably map the output landscape for the much broader parametric space. However, the neural network does occasionally fail. We found that the neural network tends to make more mistakes in clustered regions (e.g., the blank regions in Fig. 5b, where data have been deleted due to large disagreement in

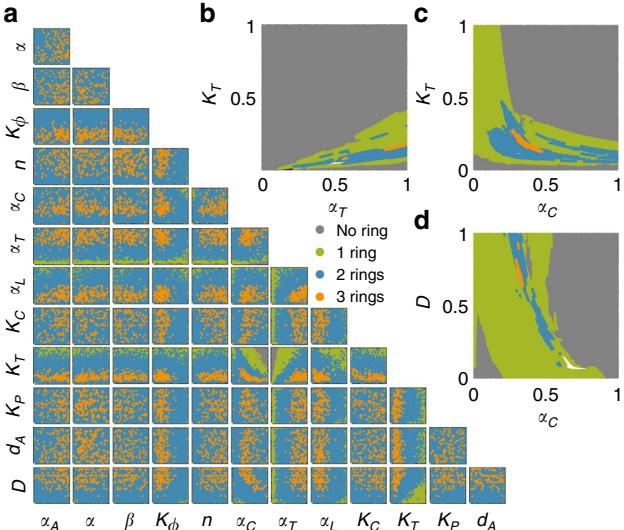

**Fig. 5** Neural network predictions enable comprehensive exploration of pattern formation dynamics. **a** Ensemble of deep neural networks enables screening through a vast parametric space. The parametric space consists of 13 parameters that were varied uniformly in the provided ranges (Supplementary Table 1). For each instance, we randomly generated all the varying parameters and used the neural network to predict the peak and distributional values for each parameter combination. We collected $10^8$ instances and discarded predictions with disagreement between ensemble predictions larger than 0.1. We then projected all the instances on all the possible 2 parameter plane. The majority of the instances generated patterns with no ring (gray), and they were distributed all over the projected parametric planes. Due to the huge number of instances, the parametric distribution of no ring (grey), one-ring (green), two-rings (blue) patterns on the projected 2D planes partially overlap. From the distribution of neural network predicted three-ring patterns (orange) over all the possible 2D parameter planes, the critical constraints to generate three-ring patterns are revealed: large domain radius ($D$), large synthesis rate of T7RNAP ($\alpha_T$), small synthesis rate of T7 lysozyme ($\alpha_L$), small half activation constant of T7RNAP ($K_T$), small half activation distance for gene expression ($K_\varphi$). The analysis also suggested correlations between $K_T$ and $\alpha_C$ (cell growth rate on agar), $K_T$ and $\alpha_T$, $D$ and $\alpha_C$. **b-d** Neural network predictions facilitate the evaluation the objective function of interest (generation of three-ring patterns). Based on the analysis above, we sought to further identify the correlation between $K_T$ and $\alpha_C$, $K_T$ and $\alpha_T$, $D$ and $\alpha_C$. For each of the screening, we varied two parameters of interest and fixed the rest. We collected $10^7$ instances and discarded predictions with disagreement between ensemble predictions larger than 0.1. We found generation of three-ring patterns requires a negative correlation between $D$ and $\alpha_C$ and a negative correlation between $K_T$ and $\alpha_C$. We also found a positive linear correlation between $K_T$ and $\alpha_T$. $\alpha_A = 0.5$, $\alpha = 0.5$, $\beta = 0.5$, $K_\emptyset = 0.3$, $n = 0.5$, $\alpha_L = 0.3$, $K_C = 0.5$, $K_P = 0.5$, $d_A = 0.5$, **b** $\alpha_C = 0.5$, $D = 1.0$. **c** $\alpha_T = 0.8$, $D = 1.0$. **d** $\alpha_T = 0.8$, $K_T = 0.3$

predictions). As such, we suspect that the neural network is more likely to fail when the system output is highly sensitive to local parameter variations, i.e., where the output landscape is rugged in the parametric space. Such a limitation could be alleviated by increasing the size of the training set. Alternatively, one could generate the training set according to local system sensitivity, such that training data are generated more densely in regions of high local sensitivity.

Our approach is generally applicable as long as each input parameter set generates a unique output (but the same output can correspond to different input parameter sets). This constraint is implied in both of our examples. In the pattern-formation circuit,

each parameter combination can generate a unique final pattern. For the stochastic model, different runs of the model will generate different levels for each molecular species (for the same parameter set). However, the distribution of these levels for a sufficiently large number of simulations is approximately deterministic—it will be deterministic for an infinite number of simulations. Therefore, each parameter set in the stochastic model leads to a unique distribution for each molecule. This constraint is satisfied in vast majority of dynamical models of biological systems, where the output can be a time series, a spatial distribution, or distribution of molecules from ensemble simulations. As such, the general framework (Fig. 1) is applicable to all these models. However, the benefit of the framework depends on the specific model of interest and the number of parameter sets to be explored. In particular, our approach is most useful when the generation of the initial data set is non-trivial but manageable. While doable, our approach will not gain much by emulating simple ODE models, which can be solved quickly. Conversely, if a model is so complex, such that, even the generation of sufficient training data could be computationally prohibitive, an alternative integration of the mechanistic model and deep learning is necessary to speed up the training process.

## Methods

**Modeling pattern formation in engineered bacteria.** The circuit consists of a mutant T7 RNA polymerase (T7RNAP) that activates its own gene expression and the expression of LuxR and LuxI. LuxI synthesizes an acyl-homoserine lactone (AHL) which can induce expression of T7 lysozyme upon binding and activating LuxR. Lysozyme inhibits T7RNAP and its transcription by forming a stable complex with it[53]. CFP and mCherry fluorescent proteins are used to report the circuit dynamics since they are co-expressed with T7RNAP and lysozyme, respectively.

The PDE model used in the current study corresponds to the hydrodynamic limit of the stochastic agent-based model from Payne et al.[33]. Because the air pocket between glass plate and dense agar is only 20 μm high, the system was modeled in two spatial dimensions and neglect vertical variations in gene expression profiles[32]. Although the PDE formulation is computationally less expensive to solve numerically than the stochastic agent-based model and better facilitates development of mechanistic insights into the patterning dynamics, it still needs a lot of computational power when extensive parameter search is needed.

The circuit dynamics can be described by the following PDEs:

$$\begin{cases} \frac{\partial C}{\partial t} = \kappa_C \Delta C + \alpha_C \frac{1}{1+\alpha T+\beta L} \cdot \frac{N}{K_N+N} C\left(1-\frac{C}{C}\right), \\ \frac{dN}{dt} = -\frac{\alpha_N}{|\Omega|}\int_\Omega C\left(1-\frac{C}{C}\right)\frac{N}{K_N+N}d\sigma, \\ \frac{dA}{dt} = \frac{\alpha_A}{|\Omega|}\int_\Omega C\frac{T}{K_T+T}\frac{K_P}{K_P+P}\varphi(x,C)d\sigma - d_A A, \\ \frac{\partial T}{\partial t} = \kappa_C \frac{VT\cdot VC}{C} - \alpha_C T\frac{N}{K_N+N}\left(1-\frac{C}{C}\right) - d_T T + \alpha_T \theta(C)\frac{T}{K_T+T}\frac{K_P}{K_P+P}\varphi(x,C) - k_1 TL + k_2 P, \\ \frac{\partial L}{\partial t} = \kappa_C \frac{VL\cdot VC}{C} - \alpha_C L\frac{N}{K_N+N}\left(1-\frac{C}{C}\right) + \alpha_L \theta(C)\frac{T}{K_T+T}\frac{A^m}{K_A^m+A^m}\varphi(x,C) - d_L L - k_1 TL + k_2 P, \\ \frac{\partial P}{\partial t} = \kappa_C \frac{VP\cdot VC}{C} - \alpha_C P\frac{N}{K_N+N}\left(1-\frac{C}{C}\right) + k_1 TL - k_2 P, \end{cases}$$
(1)

where $C(t, x)$ is the cell density; $N(t)$ is the nutrient concentration; $A(t)$ is the AHL concentration; $T(t, x), L(t, x), P(t, x)$ are cellular T7RNAP, lysozyme and the T7-lysozyme complex density, respectively. See Supplementary Table 1 for description of all model parameters.

**Modeling of MYC/E2F pathway in cell-cycle progression.** The stochastic differential equation (SDE) model use in this study is the system describing the MYC/E2F pathway in cell-cycle progression[42,43,54] (Supplementary Fig. 8a). Upon growth factor stimulation, increases in MYC lead to activation of E2F-regulated genes through two routes. First, MYC regulates expression of *Cyclin D*, which serves as the regulatory components of kinases that phosphorylate pocket proteins and disrupt their inhibitory activity. Second, MYC facilitates transcriptional induction of activator E2Fs, which activate the transcription of genes required for S phase. Expression of activator E2Fs is reinforced by two positive feedback loops. First, activator E2Fs can directly binds to their own regulatory sequences to help maintain an active transcription state, second, activator E2Fs transcriptionally upregulate CYCE, which stimulates additional phosphorylation of pocket proteins and prevents them from sequencing activator E2Fs.

To capture stochastic aspects of the Rb-E2F signaling pathway, we adopted the Chemical Langevin Formulation (CLF)[55]. We adjusted the units of the molecule concentrations and the parameters so that the molecules are expressed

in molecular numbers.

$$\frac{dX_i(t)}{dt} = \sum_{j=1}^{M} \nu_{ji} a_j[X(t)] + \sum_{j=1}^{M} \nu_{ji} a_j^{\frac{1}{2}}[X(t)]\Gamma_j(t) + \omega_i(t) \quad (2)$$

Where $X_i(t)$ represents the number of molecules of a molecular species $I$ ($i = 1,...,$ $N$) at time $t$, and $X(t) = (X_1(t), ... , X_N(t))$ is the state of the entire system at time $t$. $X_i(t)$ interact through a set of M reactions. $a_j[X(t)]dt$ ($j = 1, ..., M$) describes the probability that the $j$th reaction will occur in the next infinitesimal time interval $[t,t+dt]$, and the corresponding change in the number of individual molecules produced by $j$th reaction is described in $\nu_{ji}$. $\Gamma_j(t)$ and $\omega_i(t)$ are temporally uncorrelated, statistically independent Gaussian noises. This formulation retains the deterministic framework (the first term), and intrinsic noise (reaction-dependent) and extrinsic noise (reaction-independent). The concentration units in the deterministic model were converted to molecule numbers, so that the mean molecule number for E2F would be ~1000. We assumed a mean of 0 and variance of 5 for $\Gamma_j(t)$, and a mean of 0 and variance of 50 for $\omega_i(t)$. The resulting SDEs were implemented and solved in Matlab. Serum concentration is fixed at $[S] = 1\%$.

Twenty-four parameters of the SDE model are generated randomly at the range provided at Supplementary Table 5. The range covers almost all the possible rates that can be found in vivo. For each of the generated combination of parameters, sample $10^4$ stochastic simulations and collect the final values of all 10 variables for each of the simulation. Each of the variables are discretized into 1000 intervals. Create a kernel distribution object by fitting it to the data. Then by using Matlab function *pdf()* to get the probability density function of the distribution object, evaluated at the values in each of the discretized interval. (Matlab code can be found on GitHub, https://github.com/youlab/pattern_prediction_NN_Shangying.git).

**Benchmarking computational acceleration.** The critical foundation underlying the neural-network-mediated acceleration is as follows: for a mechanistic model to map input parameters to the system output (e.g. the spatial distribution of a molecule for the PDE model), it has to integrate the equations with sufficient high resolution and accuracy. Once trained, the neural network entirely bypasses the generation of these time courses, leading to massive acceleration.

The extent of acceleration would vary depending on specific mechanistic models and their neural network counterparts. For the pattern formation system, a PDE simulation on average takes 350 s on a laptop computer; a neural network prediction on average takes 0.012 s on the same computer, leading to ~30,000-fold acceleration. When conducting this benchmarking, we used a version of the Tensorflow without GPU support (our PDE simulator also did not incorporate GPU support).

**Computing platform and data preparation.** We used the Duke SLURM computing platform to simulate mechanism-based models and preparing data for training neural networks. We used Google cloud machine learning platform for hyper-parameter tuning. We use python 3.5 platform and implement TensorFlow 2.0 for neural network design and trainings/validations/tests. Source codes are available on github.com (https://github.com/youlab/pattern_prediction_NN_Shangying.git).

Machine learning algorithms do not perform well when the input numerical attributes have very different scales. During data preprocessing, we use min-max scaling to normalize all the input parameters to be within the region 0–1. We also extract the peak value from the distribution and log the peak value. We use LSTM network for prediction of the normalized distribution. Specifically, we divided the space along radius axis to 501 points. And each point is associated with an LSTM module (see below) for prediction.

**LSTM networks.** Most of the neural networks are feedforward neural networks, where the information flows from the input layer to the output layer. A RNN has connections pointing backwards. It will send the predicted output back to itself. An RNN when unrolled can be seen as a deep feed-forward neural network (Supplementary Fig. 2a). RNN is often used to predict time series data, such as stock prices. In autonomous driving systems, it can anticipate car trajectories and help avoid accidents.

However, the ordinary RNN cannot be used on long sequence data. The memory of the first inputs gradually fades away due to the transformations that the data goes through when traversing an RNN, some information is lost after each step. After a while, the RNN state contains virtually no trace of the first inputs[56]. To solve this problem, various types of cells with long-term memory have been introduced and the most successful/popular one is the LSTM network. Supplementary Figure 2b showed the architecture of an LSTM cell. An internal recurrence (a self-loop) is added on top of the outer recurrence of the RNN. This self-loop is responsible for memorizing long-term dependencies (See Supplementary Notes).

**Network structure.** Supplementary Figure 2d demonstrates the structure of the employed Deep LSTM network, which consists of an input layer with inputs to be the parameters of mechanism-based model, a fully connected layer, LSTM arrays, and two output layers, one for predicting peak values of distributions, one for

predicting the normalized distributions. First, the parameters of differential equations are connected to the neural network through a fully connected layer. Fully connected layer means all the inputs are connected to all the neurons in that layer. The activation function is Exponential Linear Unit (ELU) and the connection weight is initialized randomly using He initialization method[57]. It then connected to another fully connected layer with one neuron for peak value prediction. The output of the first fully connected layer is also connected to a sequence of LSTM modules for predicting distributions. We use Adam optimization algorithm to adaptive moment estimation and gradient clipping to prevent exploding gradients. (See Supplementary Notes for more details).

**Network optimization.** We used both cross entropy (tf.nn.sigmoid_cross_entropy_with_logits) and mean squared error (MSE) (tf.reduce_mean(tf.square())) for calculating the cost function of the neural networks. Cross entropy originated from information theory. If the prediction is exactly the same as the pattern from simulation, cross entropy will just be equal to the entropy of the pattern from simulation. But if the prediction has some deviation, cross entropy will be greater by an amount called the Kullback-Leibler divergence (KLD). The cross entropy between two distributions p and q is defined as $H(p, q) = -\sum_s p(s)\log q(s)$. In our study, we found using either of the cost function did not alter the accuracy of our network and the analysis.

**Network performance.** In the main text, we used RMSE to evaluate the difference between the data generated by PDE simulation and those generated by the neural network. Assuming that there are two distributions, $p$ and $q$. Each consisting n discrete points, the RMSE was calculated using the following equation: $RMSE = \sqrt{\frac{1}{n}\sum_{i=1}^{n}(p_i - q_i)^2}$. We also included $L^2$ norm and KS-distance evaluation methods to characterize the agreement between the continuous distributional data generated by PDE simulation and those generated by the neural network in the Supplementary Table 4 and Supplementary Table 6. We found that no matter which method is used, the conclusion stays the same.

**Ensemble prediction.** To make a prediction for a new instance, we need to aggregate all the predictions from all predictors. The aggregation function is typically the statistical mode (i.e., the most frequent predictions) for classification problems and the average for regression problems. Previously, there is no study on what the aggregation function shall be for distributional predictions. Here, we proposed that for distributional predictions, similarity score between different predictions are calculated. The similarity score can be RMSE, KL divergence, $R^2$, or other similarity function between two distributions. In this paper, we choose RMSE for calculating the similarity score. Each predictor is associated with a score based on the average of the RMSE of its prediction in comparison with all other predictions. The final prediction will be the one with the minimal score.

In many cases, an aggregated answer from a group of people is often better than one person's answer, which is called wisdom of the crowd[40]. Similarly, aggregating predictions of a group of predictors will often get better predictions than with only one predictor. A group of predictors is called an ensemble and this algorithm is called an ensemble method[41].

Predictions from an ensemble of neural networks has lower error than predictions from a single neural network in predicting distributional data (Supplementary Table 4). Also, the disagreement in prediction between predictors is positively associated with errors in predictions for test data set (Fig. 4b). Disagreement in prediction can be used as an estimate of the errors in predictions. In this way, we can rank our predictions with different confident levels.

**Reporting summary.** Further information on research design is available in the Nature Research Reporting Summary linked to this article.

## Data availability
The datasets generated during and/or analyzed during the study are available in the Supplementary Information.

## Code availability
The code used for data generation and/or analysis in the study are available on github.com (https://github.com/youlab/pattern_prediction_NN_Shangying.git).

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

## Acknowledgements

We thank Li Ma, Zhuojun Dai, Tatyana Sysoeva, Meidi Wang and Fred Huang for discussions and comments; Shiyuan Mei, Esther Brown and Yuanchi Ha for help editing the manuscript. Duke Compute Cluster (DCC) for assistance with high-performance computation solutions. This study was partially supported by the Office of Naval Research (L.Y.: N00014-12-1-0631), National Science Foundation (L.Y.), National Institutes of Health (L.Y.: 1R01-GM098642), and a David and Lucile Packard Fellowship (L.Y.)

## Author contributions

S.W. and L.Y. designed the research framework. S.W., K.F., C. Z., and K.A.H developed the code for training the LSTM network. S.W., N.L., and Y.C. developed the

mathematical model and matlab codes for the pattern formation circuit. S.W. developed and improved the mathematical model and matlab codes for the Myc-E2f model. S.W. and K.F. developed codes for neural network prediction and parameter screening. S.W., F.W., and L.Y. developed prediction protocol that uses an ensemble of neural networks. S.W. generated all the results presented in the paper. S.W. and L.Y. wrote the paper with inputs from other co-authors.

## Competing interests
The authors declare no competing interests.
