## [Peer Review File · Nature Communications]

Reviewers' comments:

Reviewer #1 (Remarks to the Author):

I find the manuscript very interesting, presentation accessible and the results significant and potentially interesting for a broader scientific community.

My only major comment is somewhat superficial calculations the authors use to conclude "30,000" speed-up in the computational speed. I really feel like there is apple-to-orange comparison here. The comparison is between the computing the transient solution of PDEs vs prediction of stationary profile in NN. I also think the authors compare single thread computation in one case vs parallel GPU computation in the other thread. I believe the authors should provide more rigorous estimate of CPU operations required to achieve compatible accuracy of the two approaches based on the established methodologies in computer science and applied math literature.

I would also suggest that in addition (or instead of R^2/MSE metric presented) the authors consider introducing a measure that is more suitable for the continuous function such as L^2 or L^1 or L^∞ norm to characterize the agreement between the profiles. If used, MSE should be explicitly defined with an equation. KS-distance could possibly be a better way to quantify the difference between the stochastic distributions.

Reviewer #2 (Remarks to the Author):

The authors present here a method leveraging artificial neural networks (ANNs, LSTMs in particular) to predict a one-dimensional spatial distribution and a probability density function ordinarily obtained from solving a partial differential equation and a stochastic differential equation, respectively. The ANN approach is $\sim 30,000$ times faster than solving the differential equations and this opens the possibility of sweeping parameters much faster and effectively than using the traditional approach. The authors convincingly demonstrate this by finding novel patterns in the pattern formation circuit by Cao et al tweaking the parameter inputs. In doing so they also find that the ANN results, even though generally accurate, can produce different results than the corresponding differential equations. To flag these cases, they use an ensemble approach in which different neural networks are trained in the same way but stochastic effects produce different weight distributions. When the different ANNs produce different results, the method is more likely to provide no match with the differential equation results.

Both these approaches (the use of LSTMs to predict differential equation results and the use of an ensemble approach to predict when this new approach and the traditional one may not match) are very interesting, potentially very useful and a novel application in computational biology. Hence, I think these results should be published. However, the authors must recognize and cite the previously existing work in solving differential equations with ANNs (which is only partially done), like for example:

- Lagaris, Isaac E., Aristidis Likas, and Dimitrios I. Fotiadis. "Artificial neural networks for solving ordinary and partial differential equations." *IEEE transactions on neural networks* 9.5 (1998): 987-1000.
- Baymani, Modjtaba, Asghar Kerayechian, and Sohrab Effati. "Artificial neural networks approach for solving stokes problem." *Applied Mathematics* 1.04 (2010): 288.
- Chiamonte, M. M., and M. Kiener. "Solving differential equations using neural networks." *Machine Learning Project* (2013).

I encourage the authors to do a thorough review of previous work, which is significant.

Other (minor) changes that I would demand before publications are:

1. The introduction should be less sanguine. While it is true that quantitative mathematical models of biological systems are being used more and more, their use is not widespread and one cannot

truthfully say that “Mathematical modeling has become an indispensable tool”. The authors should provide a more realistic assessment of mathematical modeling in biology. Otherwise the type of researchers that the field needs to attract (mathematicians, computer scientists, engineers, physicists...) are provided too rosy a picture of the situation and frictions with experimental biologists might ensue.

2. In page 3 (“For example, consider a model with 10 parameters...” and after), one should account for the possibility of not needing to test all parameters combinatorially, but using instead some type of Design of Experiment approach to wisely choose which (limited) parameter sets to run.

3. In page 7, it is mentioned that “When the data size is 10^5 , the MSE is sufficiently small.”. What was the criterion for that? Regarding this point, Figure 2C should be plotted in terms of relative error rather than MSE.

4. The authors should include a quantification of the exceptions discussed in page 7 (line “We further validate them by numerical simulations using the PDE model and find rare exceptions”). Is it 0.00001% or 3%? It does not have to be a perfect quantification (I understand it is hard to estimate quantify rare events), but we do need some type of quantification.

5. Why did the authors choose 4 models for the ensemble approach? (page 9, line “To test this notion, we trained four neural networks...”). We need a justification for this number.

6. In page 10, it refers to figure S7A, when it should be S8A, according to my understanding. Otherwise, please explain.

7. The predictions for RB in figure S9A are much worse than any of the others. Why? Can the authors hypothesize a reason?

Response to reviewers' comments

Reviewer #1 (Remarks to the Author):

I find the manuscript very interesting, presentation accessible and the results significant and potentially interesting for a broader scientific community.

We thank the reviewer for recognizing the relevance and significance of our work and for constructive suggestions and comments.

My only major comment is somewhat superficial calculations the authors use to conclude "30,000" speed-up in the computational speed. I really feel like there is apple-to-orange comparison here. The comparison is between the computing the transient solution of PDEs vs prediction of stationary profile in NN. I also think the authors compare single thread computation in one case vs parallel GPU computation in the other thread. I believe the authors should provide more rigorous estimate of CPU operations required to achieve compatible accuracy of the two approaches based on the established methodologies in computer science and applied math literature.

We thank the reviewer for the insightful comments.

Our estimate of $\sim 30,000$ -fold acceleration was based on the observation that: (1) on average, a PDE simulation for our model takes 350sec on a laptop computer, and (2) on average, each NN prediction takes 0.012 sec on the same computer. When conducting this benchmarking, we used a version of Tensorflow without GPU support (our PDE model also did not use GPU support).

The reviewer is completely correct in that the acceleration originates from the fact we have bypassed the generation of time courses in our NN predictions. In making these predictions, our objective is to map input parameters (e.g., kinetic rate constants) to the final spatial patterns. From this perspective, our comparison is indeed appropriate.

This point is critical foundation for the anticipated acceleration in general. We noted this point in our main text (page 5) as well:

“The key to the use of the deep learning is to establish this mapping through training to bypass the generation of time courses, leading to a massive acceleration in predictions (Figure 1)”

We also agree with the reviewer on the 2nd point. The extent of acceleration ($\sim 30,000$ fold) is specific to the models we used. It can vary significantly based on other factors (some of these were noted by reviewers). For instance, the computational efficiency of a PDE model depends not only on the model complexity and space/time discretization, but also on implementation, programming languages, hardware specification, and so on. Similar for the NN, its computational efficiency also depends on choice of underlying architecture and other implementation details.

We have revised the main text to better clarify the basis of our estimate and the empirical nature of this estimate.

I would also suggest that in addition (or instead of R^2 /MSE metric presented) the authors consider introducing a measure that is more suitable for the continuous function such as L^2 or L^1 or L^∞ norm to characterize the agreement between the profiles. If used, MSE should be explicitly defined with an

equation. KS-distance could possibly be a better way to quantify the difference between the stochastic distributions.

We thank the reviewer for this insightful suggestion, which made us realize a significant typo in our manuscript.

In our analysis, we realized that the calculations performed to calculate the difference between predicted distributions from neural network(s) and that from numerical simulations in the manuscript are RMSE, instead of MSE. The only place MSE was used is to calculate the cost function of the neural networks during training processes (line 380). We have corrected this in the revised manuscript.

In addition to RMSE and R^2 , we have now also included additional metrics (L^2 norm for PDE model, L^2 norm and KS-distance for the stochastic model) in the Supplementary Information (Tables S4 and S6) to characterize the agreement between the profiles. In general, different metrics give qualitatively consistent evaluation – a prediction judged as accurate by one is also accurate when judged by another.

Reviewer #2 (Remarks to the Author):

The authors present here a method leveraging artificial neural networks (ANNs, LSTMs in particular) to predict a one-dimensional spatial distribution and a probability density function ordinarily obtained from solving a partial differential equation and a stochastic differential equation, respectively. The ANN approach is ~30,000 times faster than solving the differential equations and this opens the possibility of sweeping parameters much faster and effectively than using the traditional approach. The authors convincingly demonstrate this by finding novel patterns in the pattern formation circuit by Cao et al tweaking the parameter inputs. In doing so they also find that the ANN results, even though generally accurate, can produce different results than the corresponding differential equations. To flag these cases, they use an ensemble approach in which different neural networks are trained in the same way but stochastic effects produce different weight distributions. When the different ANNs produce different results, the method is more likely to provide no match with the differential equation results. Both these approaches (the use of LSTMs to predict differential equation results and the use of an ensemble approach to predict when this new approach and the traditional one may not match) are very interesting, potentially very useful and a novel application in computational biology. Hence, I think these results should be published.

We thank the reviewer for finding the work interesting and novel, and for accurately summarizing our major points, and for making insightful and constructive suggestions and comments.

However, the authors must recognize and cite the previously existing work in solving differential equations with ANNs (which is only partially done), like for example:

- Lagaris, Isaac E., Aristidis Likas, and Dimitrios I. Fotiadis. "Artificial neural networks for solving ordinary and partial differential equations." *IEEE transactions on neural networks* 9.5 (1998): 987-1000.
- Baymani, Modjtaba, Asghar Kerayechian, and Sohrab Effati. "Artificial neural networks approach for solving stokes problem." *Applied Mathematics* 1.04 (2010): 288.
- Chiaramonte, M. M., and M. Kiener. "Solving differential equations using neural networks." *Machine Learning Project* (2013).

I encourage the authors to do a thorough review of previous work, which is significant.

We thank the reviewer for pointing out these relevant papers. Indeed, we should more clearly distinguish our work with these previous studies.

These studies have focused on using NNs to facilitate numerically solving differential equations, including the generation of time courses. In contrast, we used NNs to make predictions by bypassing the generation of time courses. As also noted in our response to Reviewer 1, bypassing the generation of time courses is the critical foundation for the overall acceleration in making predictions on certain (but not all) aspects of the mechanism-based models.

We have cited these papers and revised our text accordingly to clarify the differences.

Other (minor) changes that I would demand before publications are:

1. The introduction should be less sanguine. While it is true that quantitative mathematical models of biological systems are being used more and more, their use is not widespread and one cannot truthfully say that “Mathematical modeling has become an indispensable tool”. The authors should provide a more realistic assessment of mathematical modeling in biology. Otherwise the type of researchers that the field needs to attract (mathematicians, computer scientists, engineers, physicists...) are provided too rosy a picture of the situation and frictions with experimental biologists might ensue.

We thank the reviewer for this thoughtful comment. We agree and we have revised the Introduction to make it more rigorous.

2. In page 3 (“For example, consider a model with 10 parameters...” and after), one should account for the possibility of not needing to test all parameters combinatorically, but using instead some type of Design of Experiment approach to wisely choose which (limited) parameter sets to run.

We thank the reviewer for these insightful comment.

We completely agree with the reviewer that, in many circumstances, we do not need to exhaust the parametric space. The Design of Experimental approach can narrow down the most effective parameter sets to run.

Our approach is complementary to the Design of Experiment approach. For instance, even with proper Design of Experiment approach, the total computational demand for a specific model can still be large (depending on the number of parameter sets to run). If so, our approach will be useful for accelerating the predictions that are deemed necessary. Conversely, when an NN is properly trained to make fast and accurate predictions, it will alleviate the need for aggressive optimization when taking the Design of Experiment approach.

We have added some discussion in the supplementary (Line 103) to address this concern.

3. In page 7, it is mentioned that “When the data size is 10^5 , the MSE is sufficiently small”. What was the criterion for that? Regarding this point, Figure 2C should be plotted in terms of relative error rather than MSE.

This is an *ad hoc* criterion. Specifically, for our predictions, we set a threshold of RMSE value on a test dataset (consisting of 2×10^4 sets of data) to be 0.3. When the data size is 10^5 , the RMSE value is below this threshold. In general, depending on specific models and the acceptable tolerance of errors, the threshold can be set differently, which could require different data sizes for effective training.

Regarding the second question, although we trained the neural network models using different size of data (as shown in x-axis), we use the same test dataset (containing 20,000 samples that is not used for training the neural network) for estimating the performance of neural networks. Since RMSEs are calculated based on the same test dataset, the values can be compared without the need of calculating the relative error. Another reason we did not use relative error ($\Delta x/x$) since there are a lot of zero values in the distributions. In light of the reviewer’s comment, we have revised the text to make our description more specific.

4. The authors should include a quantification of the exceptions discussed in page 7 (line “We further validate them by numerical simulations using the PDE model and find rare exceptions”). Is it 0.00001% or 3%? It does not have to be a perfect quantification (I understand it is hard to estimate quantify rare events), but we do need some type of quantification.

We regret the confusion.

The number of “rare” exceptions depends on the threshold discrepancy (as evaluated by RMSE) between distributions generated by the NN and the distributions generated by the PDE model. As shown in Supplementary Table S3, if we set the threshold to 0.1, 81 out of 1284 (i.e., 6.31%) the NN predicted 3-ring patterns are not reliable.

We have revised the main text to clarify this point.

5. Why did the authors choose 4 models for the ensemble approach? (page 9, line “To test this notion, we trained four neural networks...”). We need a justification for this number.

The choice was *ad hoc*. As shown in Figure S6B, we tried 3, 4, 5 ensembles of neural networks for prediction. The ensemble predictions from the three cases were not significantly different.

We have revised the main text to clarify this point.

6. In page 10, it refers to figure S7A, when it should be S8A, according to my understanding. Otherwise, please explain.

We thank the reviewer for catching this typo. We have corrected it in the revision.

7. The predictions for RB in figure S9A are much worse than any of the others. Why? Can the authors hypothesize a reason?

We regret the lack of clarity in our presentation.

We speculate that the worse performance in predicting RB is likely due to the high sensitivity of RB distributions at certain parametric space.

We have added this speculation to the figure legend.

REVIEWERS' COMMENTS:

Reviewer #1 (Remarks to the Author):

It is somewhat regretful that authors opted out of the more detailed investigation of the factors affecting the claimed speed-up between the PDE and ANN-based prediction of stationary patterns. However, while I think that aspect would've strengthen the paper further, the paper is very strong as is and I have no further objections.

Reviewer #2 (Remarks to the Author):

The authors have addressed all my comments. I recommend publication.

REVIEWERS' COMMENTS:

Reviewer #1 (Remarks to the Author):

It is somewhat regretful that authors opted out of the more detailed investigation of the factors affecting the claimed speed-up between the PDE and ANN-based prediction of stationary patterns. However, while I think that aspect would've strengthened the paper further, the paper is very strong as is and I have no further objections.

Reviewer #2 (Remarks to the Author):

The authors have addressed all my comments. I recommend publication.

We thank both the reviewers for finding the manuscript much improved and the recommendations for publication.